# Morphogen Signals Shaping the Gastric Glands in Health and Disease

**DOI:** 10.3390/ijms23073632

**Published:** 2022-03-26

**Authors:** Claudia Zagami, Diana Papp, Alice Anna Daddi, Francesco Boccellato

**Affiliations:** Nuffield Department of Clinical Medicine, Ludwig Institute for Cancer Research, University of Oxford, Oxford OX3 7DQ, UK; claudia.zagami@ludwig.ox.ac.uk (C.Z.); diana.papp@ludwig.ox.ac.uk (D.P.); alice.daddi@ludwig.ox.ac.uk (A.A.D.)

**Keywords:** stomach, morphogen signalling, carcinogenesis, gastric glands, atrophic gastritis, intestinal metaplasia, *Helicobacter pylori*, organoids, mucosoids, stem cells

## Abstract

The adult gastric mucosa is characterised by deep invaginations of the epithelium called glands. These tissue architectural elements are maintained with the contribution of morphogen signals. Morphogens are expressed in specific areas of the tissue, and their diffusion generates gradients in the microenvironment. Cells at different positions in the gland sense a specific combination of signals that instruct them to differentiate, proliferate, regenerate, or migrate. Differentiated cells perform specific functions involved in digestion, such as the production of protective mucus and the secretion of digestive enzymes or gastric acid. Biopsies from gastric precancerous conditions usually display tissue aberrations and change the shape of the glands. Alteration of the morphogen signalling microenvironment is likely to underlie those conditions. Furthermore, genes involved in morphogen signalling pathways are found to be frequently mutated in gastric cancer. We summarise the most recent findings regarding alterations of morphogen signalling during gastric carcinogenesis, and we highlight the new stem cell technologies that are improving our understanding of the regulation of human tissue shape.

## 1. Introduction

The mucosa of the adult stomach shows a complex architecture building up from deep invaginations of the epithelium called gastric glands (Figure 1A). Early morphological studies of the mouse stomach mucosa combining radioautography and electron microscopy (EM) have shown that glands contain distinct cell types. The distribution of these cells divides the gland into four regions: base, neck, isthmus, and pit (also known as foveola). The gland base mostly contains zymogenic or chief cells (and their precursors), which secrete digestive pro-enzymes (zymogens) stored in intracellular granules. The neck and the pit are characterised by mucus-secreting neck and foveolar cells, respectively. The mucins (MUC6 in the neck and MUC5AC in the foveola) are also stored in granules, and once secreted, they form a thick barrier that protects the epithelium from toxic compounds and microbes. Acid-producing parietal cells are predominantly present in the neck region, but they are also present at the isthmus. The hormone-producing enteroendocrine cells are scattered throughout the gland, and they regulate the secretion of acid and digestive pro-enzymes by sensing the presence of food [1]. The isthmus contains granule-free undifferentiated cells, which have been identified as the progenitor of all lineages [2]. From the isthmus, cell precursors can proliferate and migrate towards the pit and give rise to pit cells or can migrate towards the neck region, becoming mucus-producing neck cells [3]. Chief and parietal cells are present mainly in the corpus, which is the central part of the stomach. These cells are less frequent or absent in the antrum (the distal part of the stomach close to the duodenum, as shown in Figure 1A) [4].

In this complex and dynamic cellular architecture, each cell performs a function in a specific position inside the gland. Morphogen signals play a fundamental role in cell fate decisions and tissue patterning. Morphogens are signalling molecules secreted by specific regions in a tissue that diffuse from the source, generating a gradient. Cells sensing these molecules activate specific transcription programmes determining their fate and position. The distribution of the morphogen predetermines the pattern of cellular response and therefore the shape (morphology) of the tissue. During embryo development, the interplay between endoderm and mesoderm drives stomach organogenesis via the activation or repression of morphogen signalling pathways such as WNT/β-catenin, Hedgehog (Hh), Bone Morphogenetic Protein (BMP), Fibroblast Growth Factor (FGF), Notch, and retinoic acid (RA) signalling pathways [5,6]. Similarly, the adult tissue architecture is also maintained by morphogen signals that instruct the cells to maintain their regenerative potential or to differentiate into a specific cell type. The standard diagnostic tool to understand human gastric pathologies is the analysis of tissue sections from biopsies taken during endoscopy. We can speculate that the histological aberrations observed in the diseased stomach (Figure 1B) are the result of an alteration in the morphogen signalling microenvironment, which modifies the proportion of cell types within the gland and therefore its shape. Specific alterations of the gland architecture are associated with a higher risk of cancer development. Cancer cells show an accumulation of somatic mutations in their genome. Most of those mutations affect genes involved in the signalling pathway of morphogens. This suggests that the changes in tissue morphology observed in cancer might originate from altered morphogen signalling.

In this review, we summarise the morphogen signals required for epithelial homeostasis in a healthy stomach. We discuss the current knowledge on the morphogen microenvironment changes in the precancerous conditions that drive tissue rearrangements leading to cancer development. Furthermore, we describe how the latest advancements in stem cell technology promoted the development of in vitro human epithelial models to study the molecular basis of tissue morphology.

## 2. Morphogen Signals Regulating Regeneration and Differentiation in the Adult Gastric Glands

The homeostasis of the gastric epithelium depends on the proliferation, differentiation, migration, and death of its cells. These events are essential to maintain the function and integrity of the epithelial barrier, and they are tightly regulated [7]. The fast turnover rate of the gastrointestinal epithelium is sustained by the replication of progenitor cells predicted to be at the isthmus in the so-called transient-amplifying compartment. These cells migrate and differentiate towards the different lineages for the renewal of the entire gland. Their rapid turnover rate is controlled by different factors such as food, hormones, and growth factors [7]. Another class of regenerative cells was observed at the base of the gland, and they are activated in case of epithelial damage [8]. These cells are long-lived, multipotent, capable of self-renewing [9], and they can enter and stay in a quiescent state of non-division [10]. Cre recombinase-based genetic tracing approaches in mice tried to define regenerative stem cells by the expression of specific marker genes. *Lgr5* [9], *Axin2* [11], *Aqp5* [12], and *Troy* [10] are targets or effectors of the WNT signalling pathway, which is usually active in regenerative cells; *Mist1* [13,14] and *Lrig1* [15] are genes known to be expressed by progenitor cells in other organs (pancreas [16], intestine [17]); *Sox2* [18] and *Bmi1* [19] are involved in endoderm development. When the Cre recombinase is driven by the promoters of these genes, the tracing marker is visible in the whole glands, suggesting that cells expressing those markers are progenitors of all lineages in the stomach. However, these genes are expressed throughout the glands even in fully differentiated cells. In particular, *Troy*, *Mist1*, and *Lrig1* were found to be expressed in differentiated chief cells [10], and *Lrig1* was also found in differentiated parietal cells [20]. These experiments are suggesting that multiple cell types are capable of regenerating and that regeneration is not a prerogative of undifferentiated cells. The marker-based tracing of stem cells was not sufficient to understand regeneration in the stomach. The presence of two distinct regenerative regions in the stomach glands of the mouse was instead confirmed by an unbiased (i.e., marker-free) lineage tracing approach, which showed that cells at the isthmus repopulate the glands at a faster pace compared to the ones belonging to the base of the gland [21]. Understanding the signals that drive regeneration and differentiation might explain how regeneration can happen from multiple cell types. The role of the microenvironment rather than the identity of the cells (defined by marker expression) might be more relevant to determine the regeneration dynamics. We focus here on the role of morphogen signalling pathways that were found to regulate healthy gland homeostasis. Table 1 summarizes the effects of morphogens in gastric epithelium homeostasis in healthy and precancerous conditions and in cancer.

### 2.1. WNT Signalling Pathway

The main morphogen pathway associated with the maintenance of regenerative capacity is the WNT signalling pathway. WNT glycoproteins can interact with multiple receptors (Frizzled and LPR5/6) to induce the translocation of the cytoplasmic β-catenin into the nucleus. β-catenin interacts with the T-cell factor/lymphoid enhancer factor (TCF/LEF) to transcribe genes involved in regeneration and proliferation [49]. The WNT signalling pathway is modulated by a complex network of co-activators and antagonists. R-spondins are secreted proteins that enhance WNT signalling by binding and stabilizing LGR4, 5, and 6 receptors. R-spondins trigger the internalization of the membrane ubiquitinase zinc and ring finger 3/ring finger 43 (ZNRF3/RNF43), preventing its association with the WNT/FZD/LRP5/6 complex. In this way, R-spondins enable the persistence of the FZD/LRP5/6 receptor complex on the cell surface, enhancing WNT signalling strength [50] [51]. Secreted frizzled proteins (FRPs) and Dickkopfs (DKKs) are instead quenching WNT ligands or blocking WNT receptors, respectively, resulting in a deactivation of the WNT pathway [52].

The first attempts to generate organoids from the human stomach revealed the importance of the WNT signalling pathway activation in the regeneration of the stomach epithelium. WNT and R-spondin are essential components of the stomach organoid cultivation cocktail, and their withdrawal causes differentiation towards foveolar cells and impaired regeneration [22,24]. The effects of R-spondin vary depending on the cell type. In mice, glands of the antrum contain two populations of stem cells with different proliferation rates, one Axin2+ Lgr5+ population localised at the gland base, and one more proliferative Axin2+ Lgr5− population localised right above the base [11]. Both these populations can give rise to differentiated cells and repopulate the antral glands, but R-spondin3 induces the hyperproliferation of the Axin2+ Lgr5− population, while the replication rate of the Lgr5+ population remains unchanged [11]. A detailed review about WNT signalling and different stem cells regulation can be found here [51]. In addition, activation of the WNT/β-catenin signalling mediated by Wnt5a was found to be also essential for regeneration occurring from Mist1+ cells after epithelial injury [25]. These experiments suggest that multiple types of cells can activate regeneration via the WNT/β-catenin signalling pathway.

In the antrum of mice, Wnt11 is the most abundant ligand, and it is found throughout the mucosa [11]. The secretion of the co-factor R-spondin3 was instead found in the stromal myofibroblasts of the muscularis mucosae near the base of the gland [11]. WNT signalling is also controlled by soluble inhibitors in the stomach: FRP1, DKK1, and DKK3 were found in cells isolated from the stroma of the lamina propria between glands [23]. The supernatant of this stroma cultivated in vitro can inhibit WNT signalling, and it can induce foveolar differentiation in co-culture with primary cells of the human stomach [23].

### 2.2. RTK/EGFR Signalling Pathway

Mitogenic peptides such as EGF, TGF-α, and amphiregulin [7] are the ligands of the Epidermal Growth Factor Receptor (EGFR). EGFR is part of a family of high-affinity receptors known as the receptor tyrosine kinase (RTK) family. Downstream mediators of EGFR (and other RTKs) include PI3K, PLC-y, RAF, ERK, and MAPK [53]. Inhibition of each of these downstream mediators might not block the pathway due to compensatory signals of the other mediators. In the stomach, EGF plays a major role in the development of the foetal epithelium [54,55], and studies in mice have shown that the activation of its receptor (Egfr) plays a role in stomach gland homeostasis, in particular on the foveolar compartment [38]. In fact, EGF has been observed mostly in the lumen of the human stomach [56]. In vitro experiments using human stem cell-driven models have shown that EGF signalling is essential for the differentiation towards the foveolar phenotype in the corpus epithelium [30]. Immunofluorescence staining and in situ hybridisation in the tissue of origin have shown that EGFR ligands EGF and TGF-α are more expressed at the foveolar and isthmus regions than in the lower part of the gland, supporting their role in controlling the differentiation towards the foveolar cells [30,57]. mTOR is a tyrosine protein kinase that can be activated by different pathways including growth factors stimulating EGFR or other RTKs. mTOR controls Lgr5+ progenitor cell homeostasis, promoting their proliferation and maintaining them in an undifferentiated state, both in homeostatic and tumorigenic conditions [58]. Furthermore, mTOR has been implicated in “paligenosis”, which is a process occurring upon injury in which differentiated cells undergo reprogramming and function as stem cells to regenerate the damaged tissue [39,40]. In the injured stomach, chief cells of the corpus show remarkable plasticity, as they can re-enter the cell cycle, act as stem cells, and repair the damaged tissue through paligenosis, restoring the tissue homeostasis [10,59,60]. Paligenosis is a conserved program that occurs in three stages involving first the repression of mTORC1 signalling, leading to autophagy and degradation of cellular elements essential for the differentiated cell functions. The second stage is the activation of stem cell-related genes such as *Mist1*, *Runx1*, *Troy*, and *Lgr5*. These genes are involved in regenerative processes as described before. Finally, the re-activation of mTORC1 enables the entry into the cell cycle and proliferation [39,60]. This process enables the stomach to boost tissue repair, but it also could increase the risk of cancer [59]. A recent review by Brown et al. gives a comprehensive insight of “paligenosis” in tissue regeneration [61]. How mTOR is activated by any morphogen or growth factor in the stomach is currently unknown.

### 2.3. TGF-β Superfamily-BMP Signalling Pathway

BMPs belong to the transforming growth factor-β (TGF-β) superfamily. When BMPs bind their receptors, they induce its phosphorylation and activate the downstream signal transducers SMAD4 and SMAD1/5/8. BMP signalling can be inhibited by small peptides such as Noggin and Chordin. In humans, BMP4 promotes foveolar cell differentiation in combination with active EGFR, while it promotes parietal cell differentiation if EGFR is inactive [30]. Accordingly, transgenic mice expressing *Noggin* in the stomach showed hyperplasia associated with reduced numbers of parietal cells [29]. The expression of *BMP*s is more pronounced at the isthmus where it promotes foveolar and parietal cell differentiation, while NOGGIN counteracts this effect from the muscularis mucosae below the glands, where it is secreted [30]. Furthermore, in the small intestine, the expression of *BMP4* represses the nuclear translocation of β-catenin, suggesting that this ligand might be detrimental for stem cells maintained by active WNT/β-catenin [62].

### 2.4. NOTCH Signalling Pathway

The NOTCH signalling pathway is another key regulator of gastric epithelial homeostasis, both in mice and humans. It is tightly regulated by direct interactions of cells expressing the receptor (NOTCH1/2/3/4) with cells expressing the ligand (DLL-1/4, JAG1/2). Activation of the NOTCH receptor results in cleavage by an ADAM (A-Disintegrin-And-Metalloprotease) family member protease and in the release of NOTCH Intracellular Domain (NICD). After nuclear translocation, NCID activates DNA binding proteins that activate NOTCH target genes. Inhibition of Notch in mice results in a reduction in the number of Lgr5+ cells as well as in a reduction in their proliferation and regenerative capacity. This reduction in stem cell features is combined with the increased expression of differentiation markers [34,35,36,41,63]. This suggests that the NOTCH signalling pathway modulates progenitor cell proliferation and differentiation in the stomach. Interestingly, this regulation is specific for the stomach antrum but not for the corpus [34,35,36,41,63]. Furthermore, inhibition of NOTCH in the stomach leads to remodelling of the epithelial cell inducing expression of intestinal markers [36].

### 2.5. Hedgehog Signalling Pathway

Hedgehog genes have three conserved homologous molecules: Sonic Hedgehog (Shh), Desert Hedgehog (Dhh), and Indian Hedgehog (Ihh). Hedgehog signals are known to play a role in controlling the proliferation, differentiation, acid secretion, and division of adult stem cells in the stomach [45]. The processing of Shh relates to pepsin A and is regulated by gastrin; therefore, it is linked to parietal cells [64]. Calcium release and the activation of Protein Kinase C (PKC) can also result in Shh secretion [65]. Hedgehog molecules travel to paracrine target cells to bind to the transmembrane receptor Patched 1 (PTCH1), thus inducing expression of their target genes through Glioma-associated oncogene homolog (GLI) family transcription factors [66]. The deletion of Shh in parietal cells (*HKCre/Shh^KO^* mice) resulted in pit hyperplasia-like glands, hypergastrinemia, and increased Ihh expression [64]. Ihh is thought to be responsible for the regulation of gastrin-induced epithelial cells in mice [67]. In addition, the absence of Shh determined a modification of the gland morphology in *HKCre/Shh^KO^* mice, which displayed loss of *E-cadherin* expression and ꞵ-catenin nuclear translocation that triggered proliferation [64]. As a consequence of pit hyperproliferation, the differentiation of the other lineages in the gland, such as chief cells, was disrupted or delayed [45]. Lastly, Shh secreted from parietal cells has been found crucial for gland regeneration after injury [46].

In summary, a combination of signals in the stomach controls the homeostasis of the gland by regulating cell regeneration, differentiation, and proliferation. It is also evident that different kinds of cells have regenerative capacity in the stomach and that a unique marker or set of markers that define stem cells does not exist. Regeneration from progenitor to differentiated cells might not be unidirectional, and a possible alternative model would rely on the concept that regeneration is an activity that almost all cell types can do depending on the instructions they receive from the microenvironment. Once they leave their stem cell niche, cells might become exposed to a different set of morphogens that promote their differentiation.

## 3. Dysregulation of Morphogen Signals in Carcinogenesis

Food with high salt content, smoke, alcohol, and especially *Helicobacter pylori* infection are the main risk factors for the development of gastric cancer. This pathogen has an extraordinary ability to colonise the epithelium of the stomach, and it contributes to cancer development [68]. The sequence of histological aberrations caused by *H. pylori* infection in humans is known as “Correa’s precancerous cascade” [69]. Chronic gastritis, atrophic gastritis, and intestinal metaplasia each have unique and well-described histological features. However, we are still unable to delineate an atlas of signals that orchestrate and drive those tissue aberrations. We review here the literature describing the role of morphogen signals in driving tissue alteration in pre-cancerous conditions.

### 3.1. Chronic Gastritis

Chronic gastritis is characterised by hyperplasia of the gastric glands that can reach a two-fold elongation compared to the original size at the antrum [70] (Figure 1B). This change in morphology is probably due to an expansion of the proliferative compartment [71]. The primary cause of chronic gastritis is the inflammation induced by *H. pylori* infection. Chronic gastritis can evolve into two mutually exclusive conditions: peptic ulcer or cancer [72,73]. The density, activity, and level of infiltration in the lamina propria of polymorphonuclear cells (mainly neutrophils) determines the severity of the inflammation. These cells produce ROS (reactive oxygen species) and proteases that damage the tissue [71]. Although this condition has been known for decades, the role of morphogen signals in driving hyperplasia has been described only recently.

Activation of the WNT pathway has been associated with stomach hyperplasia, which is a typical feature of chronic gastritis. Infection with *H. pylori* in mice is associated with an increase in gland height only when bacteria are found within the glands [26]. Axin2+ Lgr5− cells proliferate after stimulation with R-spondin3, which has been found to be increased in the stroma of the muscularis mucosae after infection [11]. A further function of the stroma is the secretion of BMP signals. The genetic ablation of BMPs in stromal fibroblasts leads to hyperproliferation and polyps in mouse models probably through a mechanism involving the activation of fibroblasts and secretion of morphogens including Wnt, Fgf, and Hgf [31]. Innate immune cells might play a relevant role in modifying the balance of morphogen signals in chronic gastritis. Granulocytes can migrate up to the neck and foveola. The degree of their intra- and peri-epithelial infiltration and their activity can predict the likelihood of progression from gastritis to other lesions [74]. It has been reported that IL-6/STAT3 signalling pathways are activated in *H. pylori*-related chronic gastritis. Both IL-6 and STAT3 interact with EGFR/ERBB signalling and are involved or linked in the regulation of proliferation and regeneration of the gastric glands [75,76,77]. The hormone gastrin has also been linked to EGFR signalling dysregulation. This hormone is secreted during food ingestion; it stimulates the secretion of hydrochloric acid from parietal cells, and it has been found to be increased during infection with *H. pylori*. In an immunodeficient mouse model of gastritis, gastrin receptor agonists were able to reduce hyperplasia [78]. Gastrin deficient (*Gas*−/−) mice show a strong imbalance of Egfr ligands, e.g., amphiregulin and epiregulin [79], corroborating the notion that Egfr might regulate proliferation of foveolar cells and hyperplasia.

The increased expansion of the proliferative compartment and the increased exfoliation of the cells on the foveolar side has been associated with activation of the EGFR signalling pathway also through TGF-α [80]. Menetrier’s disease is a disorder characterised by the expansion of the foveolar cells in the stomach corpus and fundus [42]. This condition is accompanied by an absence of parietal cells and by an enlargement of the stomach mucosa. This condition, known as foveolar hyperplasia, is related to an overproduction of TGF-α, which over-activates EGFR–PI3K–AKT signalling. The alterations seen in Menetrier’s have a different aetiology, but they have been correlated with the chronic inflammatory process in the stomach that follows the *H. pylori* infection and precedes cancer development [81].

### 3.2. Atrophic Gastritis

Gastric mucosal atrophy is morphologically defined by the loss of the typical stomach glandular structure following mucosal injury caused by *H. pylori* infection. Parietal and chief cells slowly disappear from the epithelium, causing reduced gastric acid secretion (hypochlorhydria) and consequentially impaired digestion [72,73] (Figure 1B). However, some chief cells start expressing trefoil factor 2 (TFF2) and repopulate the gland, acting as reserve stem cells [82]. This condition is associated with a reorganisation of the gastric unit. The number of proliferative cells increases at the isthmus, promoting foveolar hyperplasia. Furthermore, mature chief cells at the base of the gland are reprogrammed into proliferating cells [83]. This latter condition can be referred to as spasmolytic polypeptide-expressing metaplasia (SPEM) [84]. SPEM might be a transient alteration to induce repair after injury [85], but persistence of this condition might increase the chance of progression to cancer [86].

The connection between chief cells reprogramming and tissue repair after ulcer injury has been investigated tracing the progeny of *Mist1*-expressing cells (a marker for chief cells): Mist1+ cells of the isthmus rapidly repopulated the whole gland. Genetic ablation of the morphogen Wnt5a limited the repair after injury [25]. This confirms the importance of the activation of Wnt/β-catenin also in regeneration following injury, and it suggests that activation of this pathway induces regeneration in different cell types.

Stem cell proliferation following atrophy has been found to be orchestrated by CD44 through ERK and STAT3 axes [87]. The expression of CD44 is triggered at the isthmus by the activation of the ERK–MAPK pathway after injury (*H. pylori* infection or loss and damage of parietal cells typical of atrophy). Our study using advanced stem cell-driven culture models revealed that EGF plays a main cell fate decision role in the gastric glands [30]. High expression of this ligand is likely to underlie the condition of atrophic gastritis. The activation of the EGFR–MEK signalling pathway via EGF promotes foveolar while impairing chief and parietal cell differentiation [30]. We have found that biopsies of atrophic gastritis show elevated levels of EGF that correlate inversely with the detection of chief and parietal cell markers (PGC and ATP4B). The reduction in gland size observed in atrophic gastritis might be due to the lack of chief and parietal cells that cannot differentiate due to the elevated level of EGF [30]. In addition, the secretion of interleukin 11 (IL-11) has also been found to trigger parietal and chief cell loss in mice and foveolar hyperplasia [88]. *H. pylori* infection might be the inducer of IL-11 whose expression has been found mainly in parietal cells. The interference of IL-11 with morphogen signalling might be mediated through STAT3, which regulates the ERK–MAPK signalling [89,90].

The disproportion of foveolar cells over other cell types observed in atrophic gastritis is also the consequence of a disbalanced Shh signalling [64]. In particular, the hormone gastrin induces foveolar epithelium hyperplasia via activation of *Ihh* expression [67]. *Ihh* expression is inversely proportional to *Shh* expression, whose reduction is implicated in the disruption of chief cell differentiation [64].

### 3.3. Intestinal Metaplasia

Metaplasia in the stomach is defined by the appearance of cells with intestinal features. These metaplastic cells express the mucin MUC2 similarly to goblet cells (Figure 1B). They can be identified by Alcian–Periodic acid-Schiff staining in stomach biopsies. Intestinal metaplasia (IM) predisposes the patient to a higher risk of developing gastric cancer [69]. IM is also characterised by the aberrant presence of the marker caudal type homeobox 2 (CDX2), which is pivotal for the differentiation of basal stem cells [91,92] and a crucial regulator for intestinal identity maintenance [93]. CDX2 mRNA is expressed at a low level in the normal stomach mucosa, but its expression becomes upregulated during chronic *H. pylori* infection [94,95].

The role of morphogen signals in driving IM has been poorly investigated. The aberrant expression of CDX2 in the stomach is an accepted feature to define metaplastic cells. It has been shown that CDX2+ gastric cells also express key elements of the BMP signalling pathway [33]. Treatment with BMP2 or infection with *H. pylori* activates the signalling cascade via SMAD4 and induces upregulation of CDX2 in gastric cancer cells [32]. This suggests that BMPs are involved in the differentiation of the metaplastic cells.

In mouse transgenic models, the persistent activation of Ras in chief cells leads to the formation of both SPEM and IM in the stomach, and inhibition of the kinase Mek1/2 restores the normal mucosa [43]. This study suggests that the dysregulation of EGFR signalling can drive metaplasia and alter the morphology of the gastric mucosa. However, whether the alteration of the EGFR signalling pathway originates from an aberrant concentration of ligands or a dysregulation in the signal transduction is currently unknown. The effect of morphogen signals reshaping the tissue in pre-cancerous conditions is summarised in Table 1.

Depending on the surrounding combination of morphogens, cells are instructed to regenerate or to differentiate. Alterations of the signals modify the homeostasis and might induce metaplasia by driving cell reprogramming. An extreme change in the morphogen microenvironment might be the underlying cause of the aberrant tissue morphology observed during the gastric precancerous cascade.

## 4. Somatic Mutations Affecting Morphogen Signalling in Gastric Cancer

Gastric cancer (GC) is the fifth most common cancer and third for mortality [96]. The 5-year survival rate is low (20–40%), since it is normally diagnosed at late stages. Cancer in the stomach can be distinguished in diffuse, mixed, and intestinal type [97], based on their histological appearance and the degree of its extension to the organ. Generally, diffuse carcinomas are characterised by a wider spread inside the mucosa compared to the intestinal type [97]. Gastric cancer of the intestinal type is the last stage of the Correa cascade [69,98]; it is caused by *H. pylori* infection, and it represents 90% of all GC. By using unbiased genetic features, the cancer genome atlas categorised GC into four subtypes: Epstein–Barr virus (EBV)-positive, microsatellite instability (MSI), chromosomal instability (CIN), and genomically stable (GS). Intestinal type GC correlates with CIN type [99], it arises from intestinal metaplasia lesions in the mucosa, and it is associated with *H. pylori* infection.

In this section, we review the most frequent mutations found in GC and their possible involvement in morphogen signalling. We speculate on the possible origin of those mutations by interpreting their associated phenotype tested in animal or organoid-based models. We analysed data from the Cancer Genome Atlas (https://www.cancer.gov/tcga, 20 January 2022) showing the frequency of mutated genes found in a cohort of 433 samples of gastric cancer by using the “TCGAbiolinks” package [100]. The “maftools” package was used to create a query of the Stomach Adenocarcinoma (STAD) project in the TCGA biobank and to retrieve the information for each patient using the “muse” pipeline [101]. Genes were clustered in pathways according to the Gene Ontology. The frequency of mutation of all genes belonging to each pathway was aggregated using the function “OncogenicPathways” in the maftools package [101]. The pathways were ranked based on the frequency of their member genes found mutated in the cohort (Figure 2). In addition, the simultaneous mutation in *CDH1* and *P53* activates the WNT pathway and promotes the growth of organoids independently from R-spondin1 [27]. Indeed, *RNF43* and *ZNRF3* (which ubiquitinate and stabilise WNT receptors) are found mutated in gastric cancer (*RNF43* mutations are frequent, while *ZNRF3* mutations are rare). Cells with concurrent mutations in both can also grow independently from R-spondin1 [27]. Mutations that are enabling the autocrine production of WNT have not yet been reported.

### 4.1. WNT Pathway in GC in Gastric Cancer

Mutations in genes belonging to the WNT pathway were found in 185 of 433 patients, with a total of 66/68 genes affected (Figure 2). Among the genes within the WNT pathway, *APC*, *CHD4*, *CTNNB1*, *LRP5*, and *FZD10* are the most mutated (Table 2). Activation of the WNT/β-catenin signalling pathway is essential for cell regeneration. In effect, normal stomach organoids require WNT and R-spondin1 ligands in the cultivation cocktail for their propagation. However, some GC organoids can grow independently from WNT and R-spondin1 ligands, indicating the potential of cancer cells to escape the control of their microenvironment. This independency can be achieved by cancer cells if there is no control over β-catenin activation or if there is an autocrine production of WNT. Through the interaction with E-cadherin (CDH1), APC is responsible to counteract the WNT signalling by retaining β-catenin in the cytoplasm. It has been observed that the concomitant loss of APC and CDH1 in organoids results in activated WNT signalling and the accumulation of nuclear β-catenin [28].

### 4.2. RTK/EGFR Signalling Pathway in Gastric Cancer

The RTK/EGFR signalling pathway is the most mutated pathway among the ones analysed with 240/433 patients with a total number of 82 mutated genes out of 85 included in the pathway (Figure 2). In particular, the receptors *EGFR* and *ERBB2/3/4* and the effector *KRAS* are among the most mutated genes in gastric cancer (Table 2). To test the capacity of cancer cells with those mutations to grow independently from growth factor stimulation, organoids from gastric cancer patients were grown without any EGFR (or any other receptor tyrosine kinase) ligand in the cultivation cocktail. Normal gastric organoids are dependent on FGF10 and EGF for their growth and regeneration. Organoids with *ERBB2/3* amplification or mutation in *KRAS* are independent from the supply of EGFR ligands [27]. Interestingly, the mechanism of independence relies on the overexpression of cell autonomous ligands such as epiregulin, which was found highly expressed in some organoids [27]. To instead test any ligand-independent activation of ERBB receptors, GC organoids can be selected using a pan ERBB inhibitor. Organoids with mutation in *KRAS* and *PIK3CA* or with amplification in *FGFR2* or *MET* could survive the inhibition of ERBB. Organoids derived from genetically modified mice with the active EGFR–RAS signalling pathway were able to grow independently from Egf, Wnt, and R-spondin [44]. Furthermore, organoids carrying mutations in *KRAS* displayed an increase in ERK1/2 phosphorylation, suggesting active MAPK signalling [28]. In an epithelium infected with *H. pylori*, the activation of pro-proliferative features mediated by activation of the EGFR might present an advantage to maintain epithelial barrier integrity. Cells carrying mutations in genes involved in EGFR signalling might be selected and outcompete normal cells.

### 4.3. TGF-β Superfamily Signalling Pathway in Gastric Cancer

The TGF-β-BMP-activin pathway is affected in 7/7 genes annotated in the pathway, and it is found mutated in 67/433 patients (Figure 2). The most frequently mutated gene is *SMAD4*, which is a co-transcription factor that mediates the transcription of all TGF-superfamily ligands. Mutations are also found in genes such as *TGFBR2* and *ACVR2A/1B* and *SMAD2/3*, suggesting a possible alteration of the TGF-ꞵ and activin signalling pathway (Table 2). Interestingly, organoids with mutations in *SMAD4* and *TGFBR2* were able to proliferate even in the absence of BMP4 inhibition, which is crucial for the growth of healthy stomach organoids [27]. These mutations might be selected to tolerate a microenvironment that is enriched in BMP or TGF-ꞵ ligands [27].

### 4.4. NOTCH Pathway in Gastric Cancer

The NOTCH pathway has been found mutated in 178/433 patients and in 62/71 genes annotated in this pathway (Figure 2). It has been shown that NOTCH signalling is frequently activated in gastric cancer [102], but its role in the aetiogenesis is unclear. NOTCH activation might promote the regeneration and proliferation of GC through the potential crosstalk with the WNT and RAS/MAPK signalling pathways [102]. The second most mutated gene in GC is *ARID1A* [103]. Interestingly, *ARID1* heterozygous mutation significantly co-occurs with NOTCH pathway genes amplification, which consequently may result in cell proliferation in GC [103].

### 4.5. Hippo Pathway in Gastric Cancer

The Hippo pathway controls organ size in animals. Its main mediators, the YAP and TAZ proteins, interact with other signalling pathways including WNT and BMPs. Although the Hippo pathway is one of the most mutated in the STED cohort (221/433 patients) with 33/38 genes involved (Figure 2), currently, very little literature has been published on how the Hippo pathway contributes to the morphogenesis of GC. Among the most mutated genes, *FAT4*,*3*,*2* are the most relevant (Table 2). FAT proteins are members of the protocadherin family, which regulate planar cell polarity. They can also inhibit YAP1 signalling that regulates proliferation and differentiation in normal stomach cells [47,48]. The effect of each morphogen pathway dysregulation in GC is summarised in Table 1.

Genomic analysis of biopsies enables the computation of the most frequently mutated genes in gastric cancer patients. Although this information is useful for developing targeted therapeutic strategies [104], an investigation of the causes and the functional consequences of those mutations is still highly desired. Most of those mutations might cause a dysregulation of morphogen signals involved in cellular regeneration, proliferation, and differentiation [105]. Cells with mutations in genes involved in morphogen signalling pathways could become insensitive to their microenvironment, and they might outcompete normal cells. This clonal advantage might be a key element for cancer progression.

## 5. D Models Studying Morphogen Signalling in the Stomach

In the past two decades, the investigation of tissue homeostasis has been heavily relying on animal models. Recently, by understanding the signals involved in cell regeneration, the group of Hans Clevers developed a revolutionary cultivation method to perpetuate the growth of epithelial cells from stem cells [106]. Organoids are a new useful tool to study the biology of epithelial barriers in vitro and outside their tissue context (Table 3). With the contribution of bioengineering, stem cell-driven cultures have been developed even further by including elements of tissue-like architecture, e.g., invaginations typical of the gastrointestinal tract (crypts and glands).

### 5.1. Gastric Organoids

Organoids are long-term stem cell-driven cultures forming multicellular structures supported by an extracellular matrix and nourished with growth factors preserving cell regenerative capacity. Organoids can be generated from adult stem cells (ASCs, from primary tissue), embryonic stem cells, or induced pluripotent stem cells (iPSCs). The first gastric organoids were established from murine pyloric Lgr5+ stem cells [9]. Gastric glands were isolated from the mouse pylorus region (next to the duodenum); then, the Lgr5+ cell population was sorted and seeded in an extracellular matrix called Matrigel. Cell growth was supported with the following morphogens: Wnt3a, R-spondin1, Egf, Noggin, and Fgf10. The replicating cells formed a monolayer of columnar cells organised in the shape of a hollow sphere [9].

Organoids were also established from the human stomach without sorting the stem cells in advance [22,24]. In these organoids, different kinds of epithelial cells were found, including mucus-producing cells but also sporadic chief cells and enteroendocrine cells. This demonstrated that the multilineage differentiation capacity of the stomach epithelium could be replicated in vitro [22,24]. The different cell types self-organized into gland and pit domains [22]. Deactivation of the WNT/β-catenin signalling pathway (by removal of WNT3a and R-spondin1 from the medium of grown organoids) induced differentiation towards the foveolar cell lineage. By using organoids, it has been demonstrated that also NOTCH signalling plays a key role in maintaining gastric antral stem cells. Inhibition of NOTCH1 and NOTCH2 resulted in a reduced organoid size and increased differentiation with an emergence of corpus and intestinal marker positive cells [36]. These data indicate that active WNT/β-catenin and NOTCH signals mediate regeneration and cell differentiation.

Gastric organoids can also be derived from induced pluripotent stem cells (iPSCs) [107,108]. The formation of organoids with adult gastric cell identity was promoted after the treatment of iPSCs with factors inducing posterior foregut differentiation (Noggin, retinoic acid, and EGF). These organoids contain pit, mucus neck, and enteroendocrine cells. Sporadic chief cells are found if the WNT/β-catenin pathway is activated after posterior foregut formation, while parietal cells differentiate when BMP4 and an inhibitor of MEK (which is part of the EGFR signalling pathway) are included in the cultivation cocktail after gastric organoid formation [107,108]. These findings highlight that the same morphogen signals (BMP, WNT, EGF) that regulate embryonic stomach development can also coordinate cell differentiation in the adult gastric epithelium.

Organoids can also be co-cultured with other cell types to investigate intercellular communication. Culturing 3D organoids with myofibroblast revealed that the latter can improve the growth and viability of gastric organoids. This supports the hypothesis that myofibroblasts might provide stem cell niche factors supporting the epithelium [109]. More recently, a co-culture of gastric organoids with innervated smooth muscle cells was established from iPSCs, demonstrating that enteric neural crest cells promote the growth, patterning, and morphogenesis of gastric epithelium [110]. The signals provided by the neural crest to the epithelium are still unknown. Furthermore, gastric organoids are offering novel opportunities to study the direct impact of infections in carcinogenesis by inducing somatic mutation in morphogen signalling-related genes. Although such a study has not been carried out in stomach organoids yet, we can learn from experiments conducted in other gastrointestinal tract-derived organoids. For example, it has been shown that short-term exposure to the genotoxic colibactin-producing *pks+ Escherichia coli* bacteria causes double-strand DNA breaks and mutations in murine colon organoids. These organoids become Wnt independent and display enhanced proliferation, similar to colorectal cancer cells [111]. As WNT regulates regeneration, this result suggests that infections can induce epithelial cells to regenerate independently from their morphogen signalling microenvironment.

### 5.2. Mucosoid Cultures

We have recently developed a new cultivation system for gastric primary cells in which cells are cultivated on porous transwell inserts at the air–liquid interface. Under these conditions, cells become polarised and form an epithelial barrier with a distinct apical and basal side. Epithelial cells in these cultures can regenerate and differentiate towards all lineages, e.g., mucus-producing foveolar cells. As the cells in this format can reproduce most of the features of the gastric mucosa, they are called “mucosoid cultures”. Mucosoids can be generated both from isolated gastric glands and gastric organoids. Single-cell suspensions of these are seeded on collagen-coated polycarbonate transwell filters. After 3 days of cultivation, the air–liquid interface is initiated by removing the culturing medium from the apical side of the cells. Over the following 10 days, the cells grow in height, and transparent mucus starts to accumulate on the top of the monolayer [23]. Importantly, it is possible to isolate cells from mucosoids, reiterate the culture, and therefore expand primary cells indefinitely. Mucosoids are a robust cell culture methodology that offers a versatile tool to study cell differentiation and host–microbe interactions. Indeed, they provide the opportunity to manipulate the cell culture from both the apical and the basal side (Table 3).

Mucosoid cultures enabled the detailed analysis of the characteristic morphogenic environment that determines the differentiation of the main cell types of the gastric epithelium. Cultivation in the presence of WNT3A and R-spondin1 resulted in an enrichment of MUC6+ cells and stem cells resembling the base of the gastric gland, while the deprivation of WNT3A and R-spondin1 initiated the differentiation of MUC5AC+ foveolar cells [23]. Further analysis uncovered that gradient distribution of the EGF, BMP, and NOGGIN signals plays a fundamental role in multilineage differentiation. The presence of both EGF and BMP also facilitates foveolar MUC5AC+ cell differentiation. In the absence of EGF and BMP, the cells differentiate to PGC+ chief cells, while BMP in the absence of EGF ensures commitment to HK-ATPaseβ+ parietal cells. Furthermore, this analysis demonstrated that an elevated level of EGF is detrimental for chief and parietal cells and is probably underlying the histologic condition of atrophic gastritis [30].

### 5.3. Stem Cell-Driven Models with Tissue Architecture

Most recently, cutting-edge technologies have been developed to imitate the architecture of the gastrointestinal epithelium [112]. The extracellular matrix can be moulded to obtain the shape of the characteristic invaginations of the gastrointestinal mucosa (glands, crypts). These moulded hydrogel-based scaffolds can also be included in transwell inserts [113,114,115] and can be seeded with dissociated cells from organoids [116,117]. In this setting, WNT3A, R-spondin1, and NOGGIN gradient were able to recreate a proliferative stem cell zone at the base and a differentiated cell zone at the lumen of the invaginations [115]. Alternatively, hydrogels can be carved by laser ablation to form a tube with crypt-like structures. These tubes with crypt structures were perfused with media and seeded with dissociated cells derived from murine organoids [117]. The spontaneous homing of Sox9+ progenitor cells to the base of the crypts could be observed. These stem cells can regenerate and cover the whole crypt-like surface similarly to real intestinal crypts [118].

Bioprinting is an alternative method to generate complex epithelial structures. This methodology has been used to observe crypt formation from printed cells (from dissociated organoids) shaped into a tube [118].

Organoids and the recent advancements in stem cell technology enable the visualisation of complex patterning events in vitro, and they promise to be a relevant model to study the effect of morphogens on tissue shape. Table 3 summarises the advantages of each stem cell-driven model depending on the application.

**Table 3 ijms-23-03632-t003:** Summary of stem cell-driven in vitro models with their advantages depending on their application. “-” = none. “+, ++, +++” = minor, medium, great advantage for the specific application.

Method	Regeneration	Differentiation	Tissue Architecture	Micro-Environment	Cross-Tissue Interaction	Host-Microbe Interaction
Organoids [22,24]	*+++*	*+*	*+*	+	+	+
Mucosoid cultures [23]	*+*	*+++*	-	+	++	+++
Scaffold-supported stem cell driven models [116,117]	+	+++	+++	++	+	+++

## 6. Conclusions

Recently, single-cell RNA-sequencing analysis of the healthy and diseased mucosa revealed the presence of subpopulations of regenerative and non-regenerative cells in the stomach with an unprecedented resolution [85,119]. Analysis of the single cells in tissues across the different stages of gastric cancer progression promises to be a valuable tool to understand the occurrence of metaplasia and cancer. However, little is still known about the microenvironment that drives these processes. Infection with *H. pylori*, tissue damage, and inflammation are the main risk factors for the development of gastric cancer. How these risk factors modify the microenvironment is still poorly investigated and highly demanded. We think that mapping the morphogen ligands and receptors at consecutive disease stages will reveal the importance of the microenvironment in gastric carcinogenesis.

## Figures and Tables

**Figure 1 ijms-23-03632-f001:**
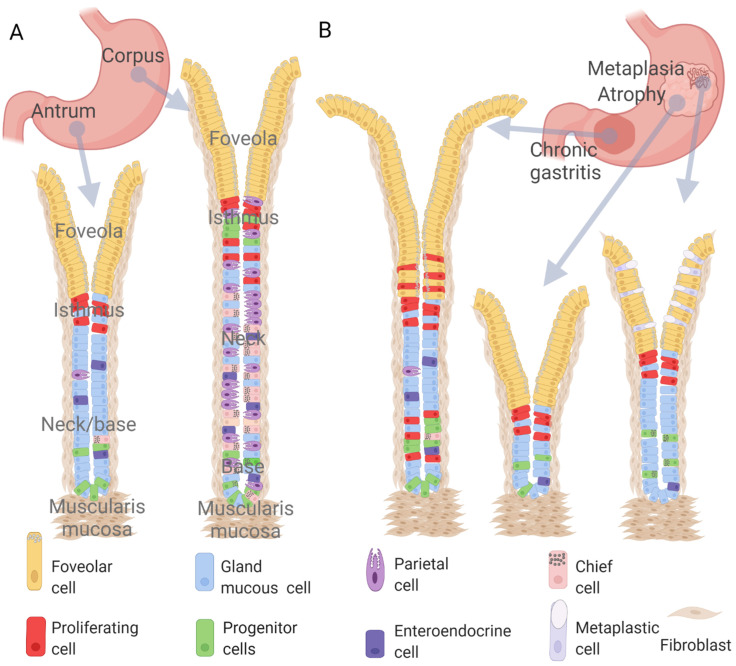
(**A**) Schematic illustration of the gastric glands from the distal and proximal part of the healthy human stomach. (**B**) Morphological changes of stomach glands in consecutive pre-cancerous conditions of the Correas’ cascade: chronic gastritis, atrophic gastritis, and intestinal metaplasia.

**Figure 2 ijms-23-03632-f002:**
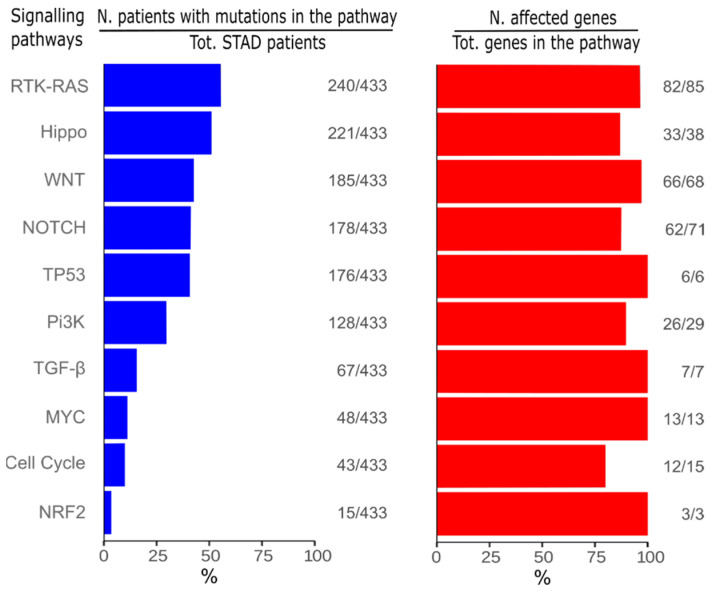
Oncogenic pathway analysis of the Stomach Adenoma (STAD) project of the TCGA biobank. Data were retrieved from the TCGA biobank using the “TCGAbiolinks” package; analysis was performed with the “maftools” package with the function “OncogenicPathways”. The pathways were ranked based on the frequency with which their member genes were found mutated in the cohort. The left part (blue bars) of the graphic represents the percentage of patients in which at least one of the genes belonging to the pathway has been found mutated. The right part (red bars) represents the percentage of genes listed in the pathway that were found mutated.

**Table 1 ijms-23-03632-t001:** Signalling pathways and their morphogenic roles in healthy tissue, in precancerous conditions, and in cancer. ↑ = increase ↓ = decreased.

Signalling Pathway	Healthy Tissue	Precancerous Conditions	Cancer
WNT	↑ Proliferation [22,23] ↓ Foveolar differentiation [22,23,24] Regeneration upon injury [25], Stem cell maintenance [9,15]	Stomach hyperplasia in chronic gastritis [26]	↑ Aberrant signalling and nuclear β-catenin accumulation (APC and CDH1 loss of function) [27,28]
BMPs	↓ Stem cell proliferation [29] ↑ Foveolar differentiation [30]	↓ Hyperproliferation and polyp formation [31] Metaplastic cell differentiation (SMAD4 and CDX2) [32,33]	↓ BMP4 proliferation regulation (SMAD4 and TGFBR2 mutations) [27]
NOTCH	↑ Stem cell maintenance and proliferation [34,35] ↓ Differentiation in the antrum [36]		↑ Proliferation (gene amplifications) [37]
EGFR	↑ Foveolar differentiation and homeostasis [38] Regulator of regeneration Paligenosis (mTOR) [39,40], Foetal epithelium development [35,41]	EGF hyperexpression foveolar differentiation (EGFR-MEK) [30], Foveolar hyperproliferation in Menetrier disease [42] ↓ Chief and parietal cell differentiation in atrophic gastritis [30], Persistent activation of RAS in chief cells in SPEM and IM [43]	↑ EGFR activation (KRAS and PIK3 mutation) (FGFR2 and MET amplification) [44]
HH	Control of proliferation [45], Control of differentiation [45], Acid secretion [45], Gland regeneration after injury (Shh) [46], Stem cell division [45]		
Hippo	Involved in proliferation and differentiation [47,48]		↓ Polarity regulation [47,48]

**Table 2 ijms-23-03632-t002:** Frequency of mutations found in specific genes in samples from the Stomach Adenocarcinoma (STAD) project of the TCGA biobank. Data were retrieved from the TCGA biobank using the “TCGAbiolinks” package. Genes are listed in the different pathways according to the “OncogenicPathways” provided by the “maftools” package of the TGCA.

RTK-RAS pathway		RAPGEF2	1.4	AXIN1	2.1	TGF-ꞵ pathway		DLL4	0.7
Gene name	mutated/	RASGRP1	1.4	FZD1	2.1	Gene name	Mutated/	FHL1	0.7
	total (%)	IRS2	1.2	FZD2	2.1		total (%)	HDAC2	0.7
ERBB4	9.7	MAP2K1	1.2	LGR4	2.1	SMAD4	5.5	NOV	0.7
ERBB3	8.3	MET	1.2	DKK1	1.8	TGFBR2	2.8	NUMBL	0.7
KRAS	6.7	RASA2	1.2	DKK2	1.8	ACVR1B	2.5	PSEN1	0.7
NF1	5.1	RASAL3	1.2	DVL2	1.8	ACVR2A	1.8	PSEN2	0.7
PLXNB1	4.2	RASGRP2	1.2	FZD7	1.8	SMAD2	1.8	SNW1	0.7
ERBB2	3.9	SHOC2	1.2	LGR5	1.8	TGFBR1	1.6	ADAM10	0.5
SCRIB	3.9	FNTB	0.9	LRP6	1.8	SMAD3	1.4	CIR1	0.5
IGF1R	3.7	RASAL1	0.9	WNT3A	1.8	NOTCH pathway		DTX3L	0.5
ROS1	3.7	RASGRP3	0.9	WNT5A	1.8	Gene name	Mutated/	EGFL7	0.5
EGFR	3.5	SHC4	0.9	PORCN	1.6		total (%)	HEY2	0.5
KSR2	3.5	SPRED3	0.9	DVL1	1.4	FBXW7	6.2	HEYL	0.5
RASA1	3.5	CBL	0.7	FZD4	1.4	CREBBP	6	PSENEN	0.5
FGFR1	3.2	ERRFI1	0.7	GSK3B	1.4	CNTN6	5.5	APH1A	0.2
RASGRF2	3.2	KSR1	0.7	LEF1	1.4	NCOR1	5.3	DTX3	0.2
IRS1	3	NRAS	0.7	WNT7A	1.4	NOTCH2	5.3	HES3	0.2
NTRK3	3	SHC1	0.7	WNT7B	1.4	SPEN	5.3	HEY1	0.2
RASAL2	3	GRB2	0.5	FZD6	1.2	NCOR2	5.1	LFNG	0.2
RET	3	MAPK3	0.5	SFRP2	1.2	NOTCH1	4.8	NUMB	0.2
ARHGAP35	2.8	MRAS	0.5	SFRP4	1.2	CNTN1	4.4	RBX1	0.2
BRAF	2.8	PPP1CA	0.5	TLE3	1.2	EP300	4.2	RFNG	0.2
FGFR4	2.8	RASGRP4	0.5	WNT10A	1.2	KDM5A	3.9	SAP30	0.2
INSR	2.8	RCE1	0.5	WNT4	1.2	NOTCH4	3.9	HIPPO pathway	
SHC3	2.8	SHC2	0.5	WNT9A	1.2	NOTCH3	3.5	Gene name	Mutated/
ALK	2.5	CBLB	0.2	AXIN2	0.9	CUL1	3.2		total (%)
FGFR2	2.5	FNTA	0.2	DKK3	0.9	JAG1	3	FAT4	19.6
NTRK2	2.5	HRAS	0.2	SFRP1	0.9	DLL1	2.5	FAT3	14.8
PDGFRB	2.5	ICMT	0.2	TCF7L1	0.9	DTX1	2.5	HMCN1	14.8
RAPGEF1	2.5	MAP2K2	0.2	TLE1	0.9	JAG2	2.5	FAT2	11.1
RASGRF1	2.5	PIN1	0.2	TLE2	0.9	THBS2	2.5	DCHS2	8.1
DAB2IP	2.3	RAC1	0.2	WNT16	0.9	KAT2B	1.8	FAT1	6.5
ERF	2.3	WNT pathway		WNT9B	0.9	NCSTN	1.8	DCHS1	5.3
INSRR	2.3	Gene name	mutated/	FZD5	0.7	RBPJ	1.8	TAOK2	4.6
FLT3	2.1		total (%)	RSPO1	0.7	CTBP2	1.6	SCRIB	3.9
JAK2	2.1	APC	7.9	TCF7	0.7	DNER	1.6	CRB1	3.7
KIT	2.1	CHD4	6.2	WNT10B	0.7	MAML1	1.4	CRB2	3
NTRK1	2.1	CTNNB1	4.6	WNT11	0.7	MAML3	1.4	LATS1	3
PDGFRA	2.1	LRP5	4.4	WNT2	0.7	RBPJL	1.4	TAOK3	3
SOS1	2.1	FZD10	3.7	WNT5B	0.7	ADAM17	1.2	LATS2	2.8
RAF1	1.8	TLE4	3.7	DKK4	0.5	CTBP1	1.2	WWC1	2.8
RASA3	1.8	AMER1	3	FRAT1	0.5	DLL3	1.2	PTPN14	2.1
SOS2	1.8	LTZR1	3	FZD3	0.5	DTX2	1.2	TAOK1	2.1
SPRED1	1.6	CHD8	2.8	FZD9	0.5	DTX4	1.2	HIPK2	1.8
SPRED2	1.6	DVL3	2.8	SFRP5	0.5	MAML2	1.2	LLGL1	1.8
ABL1	1.4	FZD8	2.8	WIF1	0.2	ARRDC1	0.9	TEAD4	1.8
ARAF	1.4	RNF43	2.8	WNT1	0.2	HDAC1	0.9	TEAD2	1.6
CBLC	1.4	WNT8B	0.5	SOST	0.5	ITCH	0.9	LLGL2	1.4
FGFR3	1.4	ZNRF3	2.8	WNT6	0.5	APH1B	0.7	CSNK1D	0.9
PTPN11	1.4	RCF7L2	2.5	WNT8A/B	0.5	DLK1	0.7	CSNK1E	0.9

## Data Availability

Not applicable.

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
