# Peer review of "Morphogen Signals Shaping the Gastric Glands in Health and Disease"

_ijms, 2022, doi:10.3390/ijms23073632_

Round 1

Reviewer 1 Report

The review titled “Morphogen signals shaping the gastric glands in health and disease” by Zagami and colleagues provides a detailed introduction to the cellular make-up of the adult stomach mucosa.  The morphogen-driven signaling pathways necessary for development and regeneration of the mucosa are discussed in detail before discussing the alterations found in these molecules and pathways in gastric cancers.  The review concludes with a discussion of the current models available to the field for study of gastric cancer biology. 

This is a well written and thorough review that will be of great value not only to the gastric biology field, but also as a general reference for anyone interested in the various signaling pathways involved in gastric mucosa development and regeneration.  The Tables and Figures provided are clear helping the reader understand the points made in the text.

Major Points –

None

Minor Points –

  1. It would be helpful to define the term morphogen for readers new to the field. It is not a term commonly used in many fields of cell or cancer biology but has a distinct meaning that is important for understanding the current manuscript.

  1. Figure 1 would benefit from labeling the four glandular regions – base, neck, isthmus, and pit. This will also help clarify which regions are affected by disease noted in 1b.

3.    Adding another model figure for Section 2 detailing the signaling pathways activated by morphogens and in what cell type/region of the stomach would help with clarity.  There is a lot of information being given and a model picture will help the reader synthesize the information provided into an overall concept.  This is highlighted by section 2.1 lines 120-136 which describes several WNT signaling pathways in various subtypes of cells that can be hard to keep straight.  Readers might get lost in the alphabet soup of signaling molecules and cell types.

Author Response

Reviewer 1

The review titled “Morphogen signals shaping the gastric glands in health and disease” by Zagami and colleagues provides a detailed introduction to the cellular make-up of the adult stomach mucosa.  The morphogen-driven signaling pathways necessary for development and regeneration of the mucosa are discussed in detail before discussing the alterations found in these molecules and pathways in gastric cancers.  The review concludes with a discussion of the current models available to the field for study of gastric cancer biology. 

This is a well written and thorough review that will be of great value not only to the gastric biology field, but also as a general reference for anyone interested in the various signaling pathways involved in gastric mucosa development and regeneration.  The Tables and Figures provided are clear helping the reader understand the points made in the text.

We appreciate that the reviewer enjoyed reading our manuscript.

Major Points –

None

Minor Points –

  1. It would be helpful to define the term morphogen for readers new to the field. It is not a term commonly used in many fields of cell or cancer biology but has a distinct meaning that is important for understanding the current manuscript.

We thank the reviewer for raising this excellent point. We have now included a definition of morphogen in the text.

“Morphogens are signalling molecules secreted by a specific region in a tissue that diffuse from the source generating a gradient. Cells sensing these molecules activate specific transcription programmes determining their fate and position.  The distribution of the morphogen pre-determine the pattern of cellular response and therefore the shape (morphology) of the tissue.”

  1. Figure 1 would benefit from labelling the four glandular regions – base, neck, isthmus, and pit. This will also help clarify which regions are affected by disease noted in 1b.

We agree and we have now labelled the different regions in figure 1

  1.   Adding another model figure for Section 2 detailing the signaling pathways activated by morphogens and in what cell type/region of the stomach would help with clarity.  There is a lot of information being given and a model picture will help the reader synthesize the information provided into an overall concept.  This is highlighted by section 2.1 lines 120-136 which describes several WNT signaling pathways in various subtypes of cells that can be hard to keep straight. Readers might get lost in the alphabet soup of signaling molecules and cell types.

Thank you for the suggestion, we also considered generating a figure for section 2, but we realised that this would be redundant with many other reviews about stem cell regulation in the stomach. If the reviewer agrees, we would just refer to other latest reviews with excellent illustrations about WNT pathway and paligenosis in the stomach. (ref 52 and ref 63).

Indeed, we hope that Table 1, which lists each signalling pathway across normal tissue, precancerous conditions and cancer will allow a comprehensive understanding of how each pathway acts in different cells types and during disease progression.

Reviewer 2 Report

In this review by Zagami et al, the authors detail the morphogens and signaling pathways they activate in different cell types of the gastric glands, with a view towards understanding the molecular pathways involved in multiple gastric diseases. The authors first carefully define the different cell types in distinct regions of the stomach, and then review the signaling pathways known to influence either the proliferation or differentiation of these cells. Understanding these pathways is critical to the next section in which they review the three types of gastric cancer. The review then comes full circle back to the pathways and mutations of each correlated with gastric cancer. A review of the literature provides important analyses of genes involved in oncogenic pathways, and informative tables are provided to indicate the key signaling pathways in healthy vs. cancerous tissue plus the frequencies of specific genes in gastric cancer. The review ends with descriptions of stem cell-based gastric models, including the most recent advancements in organoid technologies. This is a comprehensive and well-written review, the descriptions are very informative and easily digested, and the figures/tables are well-designed. There are some issues that should be considered by the authors as follows.

Figure 1, the distinct regions of the gland should be labeled with base, neck, isthmus and pit, to help differentiate the cell types of each region in the descriptions.

Lines 36-37, add "(enteroendocrine cells)" to the definition of endocrine cells.

Lines 43-44, the authors should edit their description of the "progenitors" to match the depictions in Fig 1. For example, does their description of progenitors include "proliferating cells" and "regenerative cells"? Which of these are considered stem cells? This should be clear as regeneration of the mature cells is a continual process, and certainly important to the disease states detailed later.

Lines 75-84, the authors describe the different progenitors vs. self-renewing cells in distinct regions of the gland, but how do these correlate with those shown in Fig. 1. This should be carefully pointed out, as describing where in the gland the repopulating cells reside will give the reader a better mental picture of the cellular positions in normal vs. diseased states.

Lines 88-95, while the intention of these sentences is well taken, i.e., to indicate that there is no distinct set of genes expressed strictly by regenerative cells in the gastric gland and that differentiated cells express the same markers, but why they chose this specific set of genes is confusing. This is important for subsequent sections that mention these genes, for example Lgr5, Axin2 and Mist1 in the WNT signaling section below. Also confusing is the jump from statements about regenerative cells in the isthmus vs. base, but then directly into the Cre-recombinase-based approach to marking these regenerative cells (and that no distinct marker was found).  Perhaps just a lead-in as to why these genes were selected as the researchers were trying to define the regenerative (stem) cell population vs. differentiated/functional cells would help. Some of these factors are key gastric cell receptors (Lgr5), stem cell transcription factors (Sox2) or cell cycle regulators (Bmi1), so defining them is important since several are mentioned later as defining characteristics of cells affected by the different morphogen-activated signaling pathways.

Lines 130-136, where are the stromal myofibroblasts positioned in relation to the structures and cells shown in Fig 1? This should be described, as should the lamina propria and muscularis mucosae, to give context of the descriptions of signaling factors and inhibitors described in this paragraph, and these regions described in subsequent sections.

Line 165, which stem cell genes are activated by repressed mTORC1 signaling? Do these overlap with those expressed in gland regenerative cells? This would be of interest to the other pathways described, as paligenosis is a rather unique in vivo de-differentiation process that is clearly important to the different gastric cancer types described later.

Minor issues:

Throughout the document there are spacing issues, in particular before or after inserted references, so suggest a careful edit.

Line 55, "pathway" should be plural (referring to the different signaling mechanisms).

Line 111, the beginning of the sentence "in" requires a capital first letter.

Lines 361-363, the authors might better define the three types of stomach cancers, for example to indicate how each type differs with regards to  "degree of its extension" in the stomach.

Line 395, define RSPO as the gene name for R-spondin 1. Also, is this for RSPO1?

Line 455, remove the extra "in".

Line 521-5622, this sentence should be merged with the previous paragraph.

Author Response

Reviewer 2

In this review by Zagami et al, the authors detail the morphogens and signaling pathways they activate in different cell types of the gastric glands, with a view towards understanding the molecular pathways involved in multiple gastric diseases. The authors first carefully define the different cell types in distinct regions of the stomach, and then review the signaling pathways known to influence either the proliferation or differentiation of these cells. Understanding these pathways is critical to the next section in which they review the three types of gastric cancer. The review then comes full circle back to the pathways and mutations of each correlated with gastric cancer. A review of the literature provides important analyses of genes involved in oncogenic pathways, and informative tables are provided to indicate the key signaling pathways in healthy vs. cancerous tissue plus the frequencies of specific genes in gastric cancer. The review ends with descriptions of stem cell-based gastric models, including the most recent advancements in organoid technologies. This is a comprehensive and well-written review, the descriptions are very informative and easily digested, and the figures/tables are well-designed. There are some issues that should be considered by the authors as follows.

We thank Reviewer 2 for reading and appreciating our manuscript. We found the comments and corrections very helpful to improve our review.

Figure 1, the distinct regions of the gland should be labelled with base, neck, isthmus, and pit, to help differentiate the cell types of each region in the descriptions.

Both reviewers have raised this point. We agree that better labelling of the glands in Figure 1 is necessary, so we have now labelled the different regions in Figure 1.

Lines 36-37, add "(enteroendocrine cells)" to the definition of endocrine cells.

Done

Lines 43-44, the authors should edit their description of the "progenitors" to match the depictions in Fig 1. For example, does their description of progenitors include "proliferating cells" and "regenerative cells"? Which of these are considered stem cells? This should be clear as regeneration of the mature cells is a continual process, and certainly important to the disease states detailed later.

We now called the green cells in Fig 1 “progenitor” and we now state that they proliferate and migrate. We just depicted where proliferating cells are according to ours and other stainings for proliferation markers in the stomach.  We agree that there are probably proliferating stem cells and proliferating differentiated cells (or in the process of differentiating), but we do not know where the border between these two different replicating cells types is.

Lines 75-84, the authors describe the different progenitors vs. self-renewing cells in distinct regions of the gland, but how do these correlates with those shown in Fig. 1. This should be carefully pointed out, as describing where in the gland the repopulating cells reside will give the reader a better mental picture of the cellular positions in normal vs. diseased states.

We now call the green cells in Fig 1 “progenitor cells”. Progenitor cells of different types are found both at the isthmus and at the base of the gland (as shown in Fig. 1). We also know that cells in the isthmus are more proliferative compared to the ones at the base. Therefore, we have labelled some cells at the isthmus in red and we included them in the legend as “proliferating” cells. We think that showing where proliferation in the gland is more frequent is also giving a good idea of the faster regenerative pace at the isthmus vs base of the gland.

Lines 88-95, while the intention of these sentences is well taken, i.e., to indicate that there is no distinct set of genes expressed strictly by regenerative cells in the gastric gland and that differentiated cells express the same markers, but why they chose this specific set of genes is confusing. This is important for subsequent sections that mention these genes, for example Lgr5, Axin2 and Mist1 in the WNT signaling section below. Also confusing is the jump from statements about regenerative cells in the isthmus vs. base, but then directly into the Cre-recombinase-based approach to marking these regenerative cells (and that no distinct marker was found).  Perhaps just a lead-in as to why these genes were selected as the researchers were trying to define the regenerative (stem) cell population vs. differentiated/functional cells would help. Some of these factors are key gastric cell receptors (Lgr5), stem cell transcription factors (Sox2) or cell cycle regulators (Bmi1), so defining them is important since several are mentioned later as defining characteristics of cells affected by the different morphogen-activated signaling pathways.

We have detailed in the text why were those genes selected to perform lineage tracing. We also rearranged the text to highlight the differences in the results obtained with marker-based vs marker-free tracing approaches. We conclude the paragraph with a sentence about the need to understand the microenvironment driving regeneration.

Lines 130-136, where are the stromal myofibroblasts positioned in relation to the structures and cells shown in Fig 1? This should be described, as should the lamina propria and muscularis mucosae, to give context of the descriptions of signaling factors and inhibitors described in this paragraph, and these regions described in subsequent sections.

Figure 1 now shows the fibroblast of the muscularis mucosae

Line 165, which stem cell genes are activated by repressed mTORC1 signaling? Do these overlap with those expressed in gland regenerative cells? This would be of interest to the other pathways described, as paligenosis is a rather unique in vivo de-differentiation process that is clearly important to the different gastric cancer types described later.

We now include the stem cell-related genes regulated during paligenosis (Lgr5, Troy, Mist Runx1) and indicate that they are involved in regenerative processes.

Minor issues:

Throughout the document there are spacing issues, in particular before or after inserted references, so suggest a careful edit.

Line 55, "pathway" should be plural (referring to the different signaling mechanisms). Done

Line 111, the beginning of the sentence "in" requires a capital first letter. Done

Lines 361-363, the authors might better define the three types of stomach cancers, for example to indicate how each type differs with regards to "degree of its extension" in the stomach.

Here we modified the sentence and inserted the following clarifying sentence which states that Generally, the intestinal-type carcinoma did not spread more widely in the mucosa compared to the diffuse carcinoma that is characterized by a wider spread inside the mucosa (contiguous plaque phenomenon, Collins & Gall 1952).: “Generally, diffuse carcinomas are characterised by a wider spread inside the mucosa compared to the intestinal-type.”

Line 395, define RSPO as the gene name for R-spondin 1. Also, is this for RSPO1?

Done

Line 455, remove the extra "in". Done

Line 521-5622, this sentence should be merged with the previous paragraph. Done